# Capture of Fermentation Gas from Fermentation of Grape Must

**DOI:** 10.3390/foods12030574

**Published:** 2023-01-28

**Authors:** Bozena Prusova, Jakub Humaj, Michaela Kulhankova, Michal Kumsta, Jiri Sochor, Mojmir Baron

**Affiliations:** Department of Viticulture and Enology, Mendel University in Brno, Valticka 337, 691 44 Lednice, Czech Republic

**Keywords:** must, fermentation gas, carbon dioxide, volatile aroma compounds

## Abstract

During alcoholic fermentation, a considerable amount of carbon dioxide (CO_2_) is produced, and the stream of CO_2_ can strip aromatic substances from the fermenting must. Aroma losses during fermentation can be significant and may lead to a reduction in wine quality. This study is focused on new fermentation gas capture technology. In the experiment, gas was captured during the fermentation of sauvignon blanc must. The concentration of individual volatile compounds in the fermentation gas was determined using gas chromatography, and the highest values were achieved by isoamyl acetate, isoamyl alcohol and ethyl decanoate. Ethyl dodecanoate achieved the lowest values of the investigated volatile substances. For sensory assessment, quantitative descriptive analysis (QDA) compared water carbonated with fermentation gas and water carbonated with commercial carbon dioxide for food purposes. Based on the results of this study, it can be concluded that the captured gas containing positive aromatic substances is suitable for the production of carbonated drinks in the food industry.

## 1. Introduction

Wine production consists of the conversion of sugar into alcohol and CO_2_, as the main products of yeast anaerobic metabolism. During wine fermentation, a broad range of aromatic substances are produced by yeast metabolism. Higher alcohols are present in higher concentrations, but volatile esters have the largest number of contributing molecules, responsible for the fruity and floral aroma of the resulting wine. Even small changes in these concentrations can have a large effect on the quality of the final wine [1,2,3].

Isoamyl acetate and hexyl acetate belong to the group of acetate esters that contain acetate as an acid group and alcohol derived from the metabolism of amino acids. They are produced through the condensation of yeast-derived higher alcohols with acetyl-CoA. This group of esters is important because they are produced at much higher levels and are easier to measure. Other ethyl esters contain ethanol as the alcohol group and medium-chain fatty acid as the acid group, where the acyl-CoA intermediates are esterified with ethanol-forming MCFA ethyl esters [2,4]. Important are aromatic substances with a low threshold of sensory perception, such as isoamyl acetate (30 µg/L), ethyl hexanoate (5 µg/L) and ethyl octanoate (2 µg/L) [5,6].

Volatile esters diffuse through the yeast membrane. When the excretion of acetate esters is rapid and complete, the excretion of ethyl esters decreases with increasing chain length, from 100% for ethyl hexanoate to 54–68% for ethyl octanoate and 8–17% for ethyl decanoate [7].

During fermentation, a considerable amount of CO_2_ is produced; from 100 g of glucose, 51.1 g of ethanol and 48.9 g of CO_2_ are formed [8]. The stream of carbon dioxide can strip aromatic substances from the fermenting must [9,10,11]. Aroma losses during fermentation can be significant and may lead to a reduction in wine quality. The loss of aromatics is a significant problem in the wine industry, and various technologies have been researched and developed to reduce these losses [12]. Many winemakers ferment at very low temperatures to reduce aroma losses. On the other hand, fermentation at low temperatures increases economic costs and the risk of stuck fermentations [13].

Currently there are many aroma recovery techniques, such as distillation, adsorption and solvent extraction, that are not commonly used in the wine industry. These techniques are often used at elevated temperatures, causing high energy consumption, and may use toxic solvents or require elaborate purification steps [14,15]. In recent years, there have been efforts to develop technologies that allow the separation and return of aromatic substances at lower temperatures, avoiding the harmful extraction process. Steam distillation, air stripping [16], the spinning cone column [17], supercritical carbon dioxide extraction [18], and membrane separation processes [19] were investigated for the purpose of aroma recovery [20,21,22,23].

The novel concept of this study is the capture of fermentation gas, which may have many uses, into pressurised bottles. In the wine industry, industrially produced CO_2_ is used as an inert gas, cooling medium or in the production of sparkling wines. Inert gas is most commonly used to protect wine, and additional CO_2_ is produced during fermentation and released into the atmosphere. Therefore, capturing and reusing CO_2_ would be an important improvement in sustainability as well as economic. In the food industry, CO_2_ is commonly used for saturating various carbonated beverages, including soft drinks, energy drinks and dairy products. Flavour and oral sensations are factors that contribute to the success of these products [24]. Carbonated beverages are dynamic systems in which different ingredients can influence the multimodal sensory perception [25]. 

The capture of fermentation gas is also important for the reduction of the carbon footprint. According to a report from 2016, approximately 12.7 million tonnes of CO_2_ released from fermentation processes are associated with the production of wine and spirits [26].

The advantage of fermentation gas is mainly the natural content of aromatic substances, which can sensorially enrich carbonated alcoholic and non-alcoholic beverages. In this study, the sensory impact on carbonated water was evaluated, and the aromatic profile of carbonated water at different pressures and fineness of the bubbles was determined. 

## 2. Materials and Methods

### 2.1. Design of Experiment

Sauvignon grapes from a vineyard in Lednice belonging to the Institute of Viticulture and Enology (Mendel University, Czech Republic) were used for the experiment. The grapes were harvested on 8 October 2020, and they were immediately crushed and subsequently pressed. The basic analytical parameters of the must were as follows: 23.1° Brix, 3.2 pH, 10.4 g/L total acidity and 254.2 mg/L YAN (yeast assimilable nitrogen).

After milling, the grapes were sulphurised to a value of 20 mg/L SO_2_. After pressing, the 500 L of grape must was racked into a 600 L stainless steel tank, and the prepared mixture (25 g/hL must) of active dry wine yeast BE THIOLS (Lallemand, France) was applied to the must, which was characterised by the production of fruit thiol compounds with minimal production of sulphur dioxide, hydrogen sulphide and acetaldehyde. This was followed by fermentation, which lasted 12 days. The fermentation temperature was 16 °C. 

### 2.2. Sampling

During the dynamic phase of the fermentation, samples of fermentation gases were taken directly from the fermentation tank after 96, 120, 144 and 168 h in three repetitions. After 12 days, a sample was taken from the compressed gas containers (20 L, max. 200 bar) where the gas had accumulated during the fermentation to analyse the final concentration of the fermentation gases. Capture of fermentation gas was performed from one 600 L steel tank. The buffer tank is based on the principle of an expansion tank with a rubber bag with a volume of 500 L. The purpose of the buffer tank is the trouble-free operation of the entire capture system. It is also the place of the first accumulation of fermentation gas, which is cooled here before it enters the compression unit. After filling with fermentation gas, the fuse is deactivated at a pressure of 10 bar and the fermentation gas is released into the compression unit (Recover 6, Mex, Brno, Czech Republic), which contains a condensation (filter R24 SL, Mex, Brno, Czech Republic) and a filtration unit (filters Paricer, A0, AA, AC010, AC 500, Mex, Brno, Czech Republic). The pressure is 200 bar. This system (Figure 1) was constructed based on the cooperation of Mendel University in Brno (Czech Republic) and the Mex company (Brno, Czech Republic). The sample was taken using a 2.5 mL gas-tight glass syringe and then used for analyses of selected substances on a gas chromatograph with mass detection (GC-MS).

### 2.3. Determination of Volatile Compounds on Fermentation Gas

Our attention was focused on eight important aromatic substances that are interesting for consumers due to their aroma. These were isobutyl alcohol, isoamyl acetate, isoamyl alcohol, ethyl hexanoate, 1-hexyl acetate, ethyl octanoate, ethyl decanoate and ethyl dodecanoate.

The sample taken was then manually dispensed from the injection syringe into the GC system in six seconds in splitless mode.

Instrumentation: Shimadzu GC-17A, Detector: QP-5050A, Software: Gcsolution

Separation conditions: Column: DB-WAX 30 m ×0.25 mm; 0.25 μm stationary phase (polyethylene glycol); gaseous sample injection volume: 2.5 mL; sampling period 0.2 min.; carrier gas flow (Helium): 1 mL/min (linear gas velocity 36 cm/s); injection chamber temperature: 200 °C; the initial column space temperature of 35 °C was maintained for four minutes, followed by a temperature gradient of 15 °C/min. up to a temperature of 200 °C; total analysis time was 15 min; the detector operated in SCAN mode, with an interval of 0.25 s in the range of 14–200, to obtain spectra for identification and selected ions with fragments 31, 34, 41, 43, 55, 56, 70, 88, 101 were monitored for quantification. The voltage of the detector was 1.5 kV. Individual substances were identified on the basis of MS spectra and retention times were compared to pure standards.

The measurement was carried out in three repetitions and the average values were calculated.

### 2.4. Use of a Mixture of Purified Fermentation Gases for Water Saturation

The complete fermentation gas was captured during the dynamic phase of grape must fermentation, and this mixture of fermentation gas was used to saturate the water. A high added value is the enrichment of the beverage with aromatic substances (isobutyl alcohol, isoamyl acetate, isoamyl alcohol, ethyl hexanoate, 1-hexyl acetate, ethyl octanoate, ethyl decanoate and ethyl dodecanoate), which give the beverages a specific fruity aroma.

In the experiment, two types of gases were compared: a mixture of fermentation gases captured during fermentation and industrially produced CO_2_ (Freshline, Airproducts, Czech Republic), with which water was saturated. The water in the 1 L vessel was cooled to 5 °C, connected to a pressure vessel and carbonated at 1, 2, and 3 bar. The pressure in the vessel was measured using a barometer on the pressure-reducing valve. 

### 2.5. Quantitative Descriptive Analysis (QDA)

QDA is one of the most comprehensive and informative tools used in sensory analysis. This technique can provide a complete sensory description of a grape cultivar. Quantitative descriptive analysis is based on the ability to train panel lists to measure specific attributes in a reproducible manner to yield a comprehensive quantitative description amenable to statistical analysis [27]. 

Altogether, seven experts participated in the sensory evaluation. The tasting room was designed to conduct sensory analyses in known and controlled conditions as described in the ISO 8589 standard. The samples were presented to the panel blind, coded. The samples were blind tasted in clear INAO glasses by qualified assessors (in accordance with the standards ISO 8586). The average of all the final ratings of a same jury for each sample was calculated.

The following sensory properties were evaluated: (i) basic tastes, (ii) fineness of the bubbles and (iii) aromatic profile. For this purpose, a 10-point scale was compiled for (i) individual flavours, (ii) degree of effervescence and (iii) evaluation of aromatic substances, and the results were recorded in overview graphs.

The results of the evaluation were statistically processed, graphs were generated for clarity and then the data were subjected to analysis of variance and other multivariate statistical techniques. 

### 2.6. Statistical Evaluation 

After the measurements, the results were processed using Microsoft Excel (Microsoft 365^®^) and statistically evaluated using the software Statistica 12 (StatSoft CR s.r.o.). Analysis of variance (ANOVA) and factorial ANOVA methods are commonly used techniques for the comparison of the classes of samples based on different features (in our case, the values of the sensory attributes). The ANOVA method with the use of post hoc tests–computed after the primary analysis–makes a pairwise comparison of the average values of the different groups of samples.

## 3. Results and Discussion

### 3.1. Determination of the Content of Volatile Substances in the Mixture of Fermentation Gases during the Fermentation Process

The concentration of volatile substances in the mixture of fermentation gas is not constant during fermentation and their production by yeast metabolism depends on their dynamics. These include factors such as external conditions (especially temperature and size and type of vessel), matrix parameters (especially pH, sugar content, amount of nitrogenous substances, amount of xenobiotics and alcohol content) and microflora composition (types of microorganisms and their amount) [10]. 

The highest levels of volatile compounds are produced during the yeast growth phase. Although the aromatic complexity during wine fermentation increases over time, the aromatic fractions produced close to the end of fermentation may have a negative effect on wine quality due to the higher concentration of volatile sulfur compounds [28,29,30].

Figure 2 shows the concentrations of the individual volatile substances contained in the fermentation gas. Concentrations were calculated from the initial concentration of reducing sugars and average pressure of fermentation gas. The first sample collection took place 96 h after the inoculation of the must and the last after 168 h to avoid the capture of undesirable substances formed in the last stage of fermentation.

Vas et al. observed that acetate esters reached their maximum concentration later in fermentation (approximately 300 h after inoculation) than the ethyl esters (200–250 h after inoculation) [31].

The formation of esters during fermentation in wine by the SPME method was observed in a study by Viana and Ebeler (2001). Isoamyl acetate and hexyl acetate production in our study followed very similar patterns throughout the fermentation. The higher concentration of hexyl acetate was observed in the earlier stage of fermentation (120 h after inoculation), and the higher concentration of isoamyl acetate was observed later (144 h after inoculation). It is important to note the different course of fermentation in the mentioned study, where the fermentation process lasted more days [32]. Viana and Ebeler observed the higher concentrations of ethyl hexanoate and ethyl octanoate in wine 150 and 300 h after inoculation. 

Muller et al. (1993) measured esters and higher alcohols in fermentation gases captured in charcoal adsorption traps for the aroma recovery purpose. The highest concentration of isoamyl alcohol was observed on the seventh day of fermentation (1200 ppm) and isobutyl alcohol on the fifth and seventh day of fermentation in white wine, and the highest loss of isoamyl alcohol was observed on the second day of fermentation in red wine [21].

In a more precise study by Morakul et al. (2013), volatile compound concentrations and their rates of change and losses in the exhausted gas were determined throughout the fermentation. Negligible amounts of isobutanol were lost, regardless of the fermentation temperature. In contrast, 56% of the ethyl hexanoate and 34% of the isoamyl acetate were stripped by CO_2_ when the temperature profile simulated red wine-making conditions. Even at a moderate temperature of 20 °C, typical of white wine fermentations, 40% of the ethyl hexanoate and 21% of the isoamyl acetate were lost [10]. 

### 3.2. Determination of the Content of Volatile Substances in the Final Mixture of Fermentation Gases 

The formation of volatile substances and their release into the fermentation gas depends on many factors, such as volatile precursors in must, yeast strain and fermentation conditions. Figure 3 shows a concentration of six important substances that are of sensory interest to consumers, captured during the dynamic phase of fermentation. Isoamyl acetate, isobutyl alcohol, 1-hexyl acetate, and ethyl hexanoate are substances that are considered to be the main components that give wine a fruity aroma; isobutyl alcohol is also significantly present in the fermentation gas mixture [2,10,28,33].

Figure 3 shows the concentrations of significant aromatic substances in the final mixture of fermentation gases–the complete fermentation gas captured during dynamic phase of grape must fermentation. The higher concentration (7.49 µg/L) in the final mixture of fermentation gases had isoamyl acetate, a substance that is characterised by a banana aroma. The second most common substance was isoamyl alcohol with a smell like a sweet aroma (5.79 µg/L) and ethyl decanoate (3.81 µg/L). According to the study by Morakul et al. (2013), isoamyl acetate and ethyl hexanoate were the most released volatiles from fermenting wine at moderate temperature.

During alcoholic fermentation, the gas–liquid transfer of volatile compounds can also be affected by the composition of the liquid phase (matrix effect) and CO_2_ release (stripping effect). The high sugar level at the beginning of the fermentation may induce the ‘salting out’ of volatile compounds, whereas the accumulating ethanol increases their solubility and thereby decreases their volatility [34,35,36,37,38]. 

In a study by Mouret et al. (2012), the production kinetics of 16 volatile carbon compounds corresponding to the predominant higher alcohols and esters produced during the alcoholic fermentation of wine were monitored using an online GC system. The gas–liquid partitioning of isobutanol, isoamyl acetate, and ethyl hexanoate was studied and showed that CO_2_ stripping had no impact on the partition coefficient (ki). Losses in the off-gas were highly compound-dependent. They were negligible for higher alcohols but very high for esters, with losses of up to 70% for ethyl hexanoate at 30 °C. Loss rates were maximal at the end of fermentation, indicating that high final temperatures, although helpful in avoiding sluggish fermentations, can be very detrimental to aroma losses [9].

### 3.3. Sensory Evaluation of Water Saturated by Fermentation Gas and in Industrially Produced CO_2_

Within the sensory evaluation of basic tastes, five tastes were monitored, namely sweet, savoury, sour, bitter and ferrous. For saturation variants were made with one, two, and three bars.

The sweet taste is dominant in the variants of saturated mixtures of purified fermentation gases; on the contrary, in industrially produced CO_2_ there is a significant sourness (Figure 4). To a lesser extent, bitter and ferrous tastes were observed, which were more pronounced in variants that were saturated with industrially produced CO_2_. These differences were due to the admixture of substances that are in the fermentation gases. All these tastes are affected by the quality of the carbonated medium (water) [22,23].

Carbonated beverages are dynamic systems in which different ingredients can influence the multimodal sensory perception [25]. According to psychophysical and neurobiological studies, carbonation in the mouth is of chemogenic origin. When the beverage is in the mouth, carbon dioxide is converted into carbonic acid by carbonic anhydrase activity, allowing carbonic acid to react with the tongue. Carbonic acid initiates a trigeminal sensation of oral irritation [39]. Recent studies have shown that the carbonation increased sour taste perception and decreased sweet taste perception [24,40]. Carbonation can also enhance savouriness and may inhibit other tastes or flavours [41].

### 3.4. Sensory Determination of Fineness Sparkling in Water Saturated with Fermentation Gas

The aim of this experiment was to assess the fineness of sparkling in water saturated with fermentation gas, as opposed to a control experiment where industrially produced CO_2_ was used to saturate the water.

The resulting graphs (Figure 5) show the fineness of pearling in a mixture of purified fermentation gases and in industrially produced CO_2_. Tasters rated fineness of beading on a scale of 1–10, with grade 10 representing fine beading and grade 1 representing aggressive coarse beading. The saturation technology is the same in both variants. It is evident that the fermentation gas mixture possesses smaller and more sensorially pleasing bubbles than the commonly used food grade CO_2_. The difference is due to the admixture of other substances in the fermentation gases and the composition of industrially produced CO_2_.

The transition of CO_2_ from the beverage to a separate bubble and its stability is related to the surface tension in the liquid. As the surface tension of the liquid increases, the internal pressure of the bubble increases, which in turn decreases with increasing bubble radius. The ambient pressure in a fluid tends to be higher than the surface tension of the fluid [42].

Differences in beading can be affected by the composition of the liquid or gas.

Stable undissolved gas cores can be hidden in submicroscopic cavities in the non-wetting surface of the vessel or in microscopically small particles suspended in the liquid. Free bubbles are released in a few hours, but small solid particles can form stable bubbles over time. The bubbles may contain carbon dioxide gas mixed with water vapor and the internal pressure of the bubble may be higher or lower due to the different capillary pressure in the cavities of the particles dispersed in the surrounding fluid [43].

An important parameter for the description of beading is the criterion indicating the ability of the microbubbles to leave the solid surface and ascend to the surface. The force required to tear bubbles depends on the surface tension at the solid-gas-liquid interface, together with the physical and chemical composition of the solids. Based on this, the microbubbles can be separated at the bottom or in the neck of the bottle [44].

The mixture of purified fermentation gases, in contrast to industrially produced CO_2_, has significantly finer and more pleasant sparkling in all three saturation batches. This fact may be caused by the different composition of impurities in the mixture of fermentation gases and industrially produced CO_2_. As a result, water carbonated with a mixture of fermentation gases has more sensory properties than water carbonated with industrially produced CO_2_. 

Industrially produced CO_2_ contains >99.9 % vol. of CO_2_, <2.5 ppm of NH4, <10 ppm of CO, <2.5 ppm of NOx and NO32-, <10 ppm of non-volatile impurities, <30 ppm of O_2_, <20 ppm of water, <0.02 ppm of aromatic hydrocarbons, <20 ppm of low-molecule hydrocarbons and <0.1 ppm of sulphur (Freshline, Airproducts, Czech Republic). 

### 3.5. Sensory Evaluation of Carbonated Water at Different Pressures and Temperatures 

In samples of carbonated water created by fermentation gas, attention was focused on the aromatic profile of the evaluated samples. The results are shown in graphs (Figure 6), and the descriptors were selected on the basis of laboratory analysis of the majority of aromatic substances in the mixture of fermentation gases (Figure 2).

The aromatic profile was established with reference to the analytical composition of the major aromatic substances in the fermentation gas mixture, namely isobutyl alcohol, isoamyl acetate, isoamyl alcohol, ethyl hexanoate, 1-hexyl acetate, ethyl octanoate, ethyl decanoate and ethyl dodecanoate, which are characterised by specific aromas.

The following results show the composition of the aromatic profile of a mixture of fermentation gases. In addition to the different degree of gas saturation (one, two and three bars), the evaluation took place at different temperatures (5, 12 and 20 °C, all at a pressure of two bars).

No significant differences were found within the pressure and temperature range, although in the 20 °C variant the smell of bananas was the most intense.

When carbonation is too low or too high, an overall unbalanced flavour results [45]. In some types of beverage, carbonation can also improve colour, body taste, flavour and overall acceptability [46]. In carbonated water, the perceived bubble size and bubble sound increased as temperature increased. Cooling, bite, burn and numbing decreased when temperature decreased [47]. Carbonation intensity was perceived to be higher at lower temperatures for both trained and untrained panelists [41,48].

## 4. Conclusions

In this study, fermentation gas was captured and the concentration of individual volatile compounds was analysed during fermentation and in the complete fermentation gas captured during the dynamic phase of grape must fermentation.

The higher concentration (7.49 µg/L) in the final mixture of fermentation gases has isoamyl acetate, isoamyl alcohol with (5.79 µg/L) and ethyl decanoate (3.81 µg/L). 

Since the mixture of fermentation gas is naturally enriched with aromatic substances that were released into the gas during fermentation, it was further used for saturating water. The basic taste of the water, the fineness of the sparkling at different carbonation pressures were assessed sensorially. Water carbonated with fermentation gas had a statically significantly more pronounced sweet taste, compared to water that was carbonated using industrial CO_2_. Water carbonated with industrial CO_2_ had a more pronounced sour taste.

Water carbonated with fermentation gas had a higher fineness of sparkling, which was most pronounced when carbonated at a lower pressure of 1 bar.

The influence of temperature and pressure on the aromatic properties of water carbonated with the fermentation gas was also monitored, but in this case no statistically significant differences were found.

Based on the results of this study, it can be concluded that the fermentation gas capture method has a future for use in the food industry, especially for production of carbonated beverages. 

## Figures and Tables

**Figure 1 foods-12-00574-f001:**
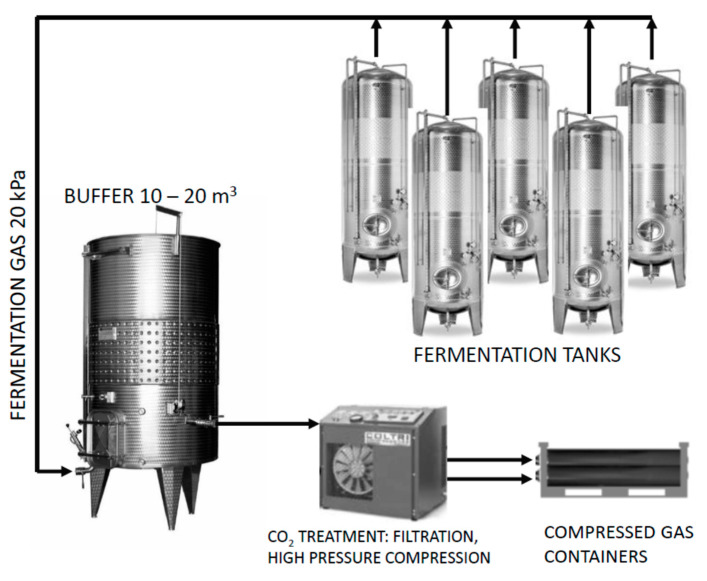
Scheme of the fermentation gas collection system.

**Figure 2 foods-12-00574-f002:**
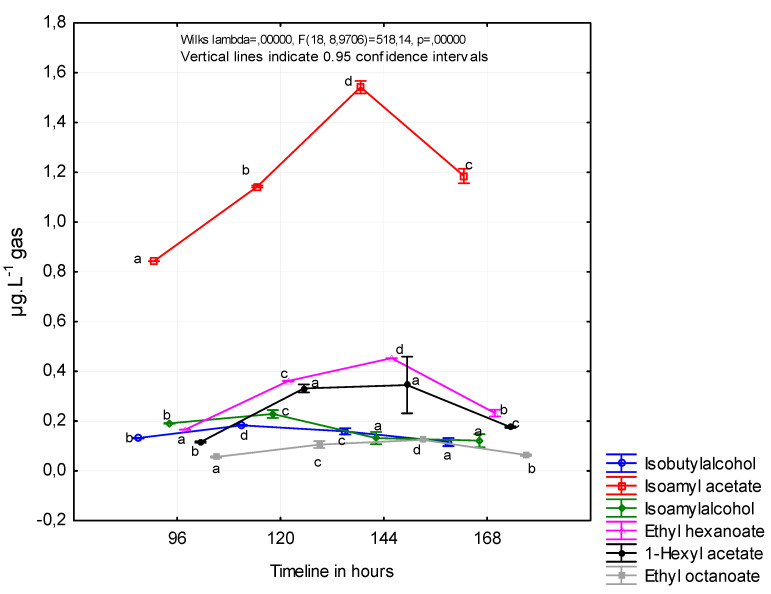
The concentrations of the individual volatile substances contained in the fermentation gas. The division into homogeneous groups (a,b,c,d) was based on the least significant difference (LSD) test, the significance level is α = 0.005.

**Figure 3 foods-12-00574-f003:**
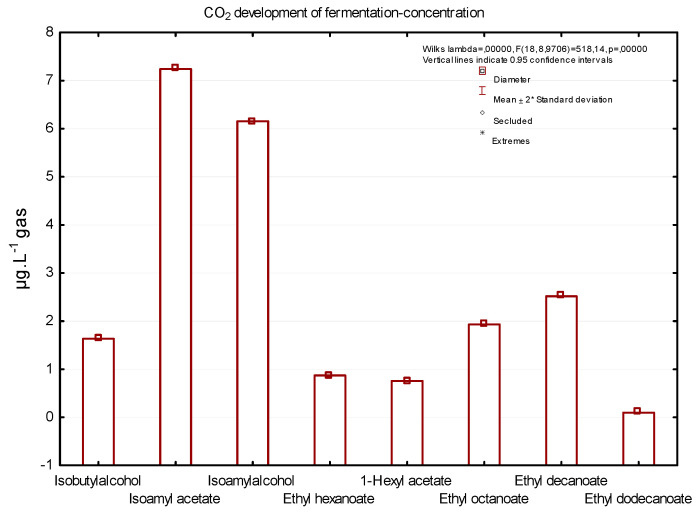
The concentrations of significant aromatic substances in the final mixture of fermentation gases.

**Figure 4 foods-12-00574-f004:**
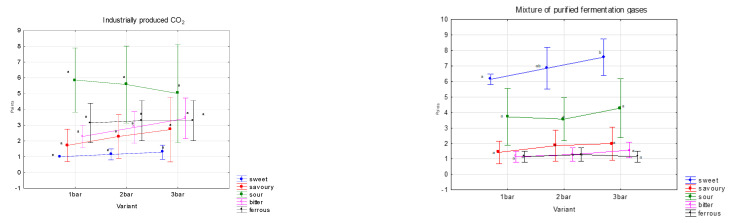
Results of sensory analysis of aromatic profile of saturated waters using fermentation gases captured from grape must fermentation and industrial CO_2_. The division into homogeneous groups (a,ab) was based on the LSD test, the significance level is α = 0.005.

**Figure 5 foods-12-00574-f005:**
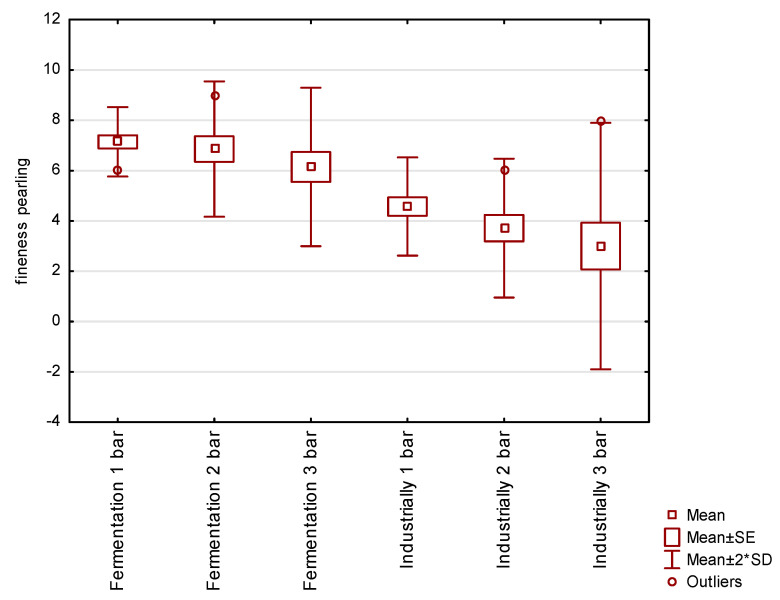
Pearling fineness of a carbonated water.

**Figure 6 foods-12-00574-f006:**
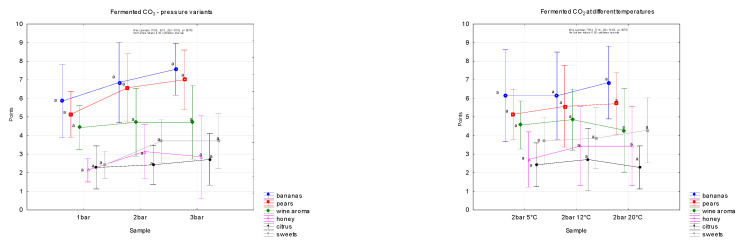
Aromatic graph of sensory evaluation of carbonated waters with mixtures of purified fermentation gases at different saturation variants (1, 2, 3 bar) and at different temperatures (5, 12, 20 ° C, all at a pressure of 2 bar). The division into homogeneous groups (a) was based on the LSD test, the significance level is α = 0.005.

## Data Availability

All related data and methods are presented in this paper. Additional inquiries should be addressed to the corresponding author.

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
