# Peer review of "Capture of Fermentation Gas from Fermentation of Grape Must"

_foods, 2023, doi:10.3390/foods12030574_

Round 1
Reviewer 1 Report
The idea is good, the results could have potential application in industry, but the execution of the study is doubtful. The main drawback is that many technical details were not reported (and they should) and statistical analysis was not done properly, which means the conclusions should be re-assessed.
L7-16: I suggest to make it clearer what is the purpose of capturing CO2 containing aroma volatiles – use for the production of various sparkling drinks and aromatization, and NOT recovery into wine, i.e. fermentation tanks.
L26-32: Authors can add information about their odor perception thresholds which clearly show their importance (especially isoamyl acetate)
L30: Maybe you meant: „Other, ethyl esters…“ (with a comma), because you were writing about acetates, not ethyl esters.
L45-52: It should be clear that these processes were not necessary related (mostly not, in fact) to wine industry.
L73-74: Abbreviations such as MN and YAN should be explained when first mentioned in the text
L75-79: Was fermentation performed without replicates? What was the volume of the grape must (percentage of the tank filled with must)? Temperature of fermentation? Yeast supplements? Sulfiting?
L80-86 & Figure 1: The process and technology of sampling should be explained in much more details. There are a number of tanks on the figure, how many replicates were performed? What is the purpose of the buffer tank – explain what it contains and how it works? CO2 treatment – filtration and compression – please describe and explain how, who constructed the device and how it works with all the parameters and details.
L90: Was GC-MS analysis performed in replicates?
L92-93: Isoamyl, not aso-amyl alcohol.
L100: Helia? Did you mean helium?
L103-105: Do you mean the scan was performed to obtain spectra for identification and selected ions were monitored for quantification?
L106: Retention times were compared to pure standards and/or linear retention indexes were calculated? Please report and describe.
L108: Which mixture – the complete gas that was captured during the dynamic phase of fermentation?
L107-135: How were the bottles filled with the gases (explain the technology) and how the final pressure in the bottles/CO2 concentration was measured (explain the measurements)? One bar, 2 bars, and 3 bars – explain the variants in more detail. Please add more details on sensory analysis, were the samples coded, how many replicates, how the assessors were trained, describe the room, etc.
L153-156: Which compounds are formed in the last phase of fermentation and have a negative effect on wine quality?
Figure 2: You measured the concentration of volatiles per liter of CO2 gas – have you measured total gas volume produced during this fermentation and how? Or you can calculate it from the initial concentration of reducing sugars and average pressure? It would be interesting to assess what was the total loss of volatile aroma compounds during this fermentation process. There are error bars – which replicates were used for this (fermentation – I suppose not, or analysis)?
L175-176: Not completely true, please be careful when discussing you results.
L194: Figure 2 – it was already stated in Line 157.
L195-198: Now you mention some completely different volatile aromas that are also important (ethyl acetate, isobutyl acetate, 2-phenethyl acetate), I agree. Why are they mentioned? Why they were not analyzed?
L200. Again, please define clearly what is the final mixture of fermentation gases.
L199-205 & Figure 3: The concentrations you mention in the text and those on Figure 3 do not correspond at all, please revise. Please use English names for the compounds in Figure 3.
Paragraph 3.3. How the statistics were performed, what were the replicates in sensory analysis results?
L224-225 & Figures: salty vs. savoury – please adjust the text and figures attributes to match. The titles should be updates so readers can understand that fermentation gases were produced during production of wine. You use sour and then acidity – please unify as well.
Authors cannot discuss the differences between waters saturated with industrially produced CO2 and mixture of purified fermentation gas since they were not statistically compared – please revise! As well, judging from very large error bars, there were no statistically significant differences between 1 bar, 2 bars and 3 bars.
L234-236: Please support these theses with some scientific evidence (at least by referencing other works).
L249: This is not Figure 2.
Figures: Please indicate statistical significance in all figures!
Figure 5 title should be more informative. What is K-mixture, what is R-industrially? (K- and R-?).
Paragraph 3.5. Authors discuss differences but it is not clear which one are statistically significant. Please indicate statistical significance, as suggested above. How did you relate volatile compounds with these grades for sensory descriptors (e.g. L303-304), it should have been done more systematically, you just guessed it approximately. Please take another approach and re-assess.
The whole Results and discussion section should be revised, first the results should be statistically elaborated in a more systematic and accurate manner and then an appropriate discussion should follow. The current version of discussion is not at the level of Foods.
The conclusion section should also be revised, there are some parts repeated from the Introduction which should be excluded. Conclusions should contain main findings of the study (in short), future prospects and directions in this research area.
English needs polishing.
Author Response
Dear Reviewer, thank you for reviewing the article and valuable comments. We have accepted all suggested corrections.
The idea is good, the results could have potential application in industry, but the execution of the study is doubtful. The main drawback is that many technical details were not reported (and they should) and statistical analysis was not done properly, which means the conclusions should be re-assessed.
We have partially changed the statistical evaluation and I have also modified some graphs. As for the conclusion, it is completely rewritten.
L7-16: I suggest to make it clearer what is the purpose of capturing CO2 containing aroma volatiles – use for the production of various sparkling drinks and aromatization, and NOT recovery into wine, i.e. fermentation tanks.
Information is added to the final part of abstract.
L26-32: Authors can add information about their odor perception thresholds which clearly show their importance (especially isoamyl acetate)
Information is added.
L30: Maybe you meant: „Other, ethyl esters…“ (with a comma), because you were writing about acetates, not ethyl esters.
Thank you, comma is added.
L45-52: It should be clear that these processes were not necessary related (mostly not, in fact) to wine industry.
Yes, thank you, we have corrected this information.
L73-74: Abbreviations such as MN and YAN should be explained when first mentioned in the text
NM was calculated to more common Brix and YAN is explained.
L75-79: Was fermentation performed without replicates? What was the volume of the grape must (percentage of the tank filled with must)? Temperature of fermentation? Yeast supplements? Sulfiting?
Yes the fermentation was done without repetition, because it was a pilot experiment. However, we did other experiments where the results were similar on other varieties and will be the subject of further studies (we will also monitor other parameters and also adsorb unwanted H2S). The volume of fermented media was 500l and was stored in a 600l stainless steel tank. The fermentation temperature was 16 °C. After milling, the grapes were sulphurised to a value of 20 mg/l free SO2. This information were added to material and method section.
L80-86 & Figure 1: The process and technology of sampling should be explained in much more details. There are a number of tanks on the figure, how many replicates were performed? What is the purpose of the buffer tank – explain what it contains and how it works? CO2 treatment – filtration and compression – please describe and explain how, who constructed the device and how it works with all the parameters and details.
Informations were added to material and methods. The samples were taken from one tank. Figure with more tanks is only for illustration, that fermentation gas can be captured from more tanks in one moment – so we need only one device for capture of fermentation gas. Each sampling was done in three repetitions. The buffer tank is because of the smooth running of the whole system. Here the gas is first accumulated and cooled so that it can then go to the compression unit. The filter is still under development and it is planned that not every gas will be filtered, but only those that need it. The compression unit compresses the fermentation gases and pushes them into pressure vessels. The pressure is 200 bar. This system was constructed in cooperation with Mendel University in Brno together with MEX.
L90: Was GC-MS analysis performed in replicates?
Yes, information was added.
L92-93: Isoamyl, not aso-amyl alcohol.
Thank you, corrected.
L100: Helia? Did you mean helium?
Thank you, corrected.
L103-105: Do you mean the scan was performed to obtain spectra for identification and selected ions were monitored for quantification?
Thank you, corrected.
L106: Retention times were compared to pure standards and/or linear retention indexes were calculated? Please report and describe.
Compared to pure standards, corrected.
L108: Which mixture – the complete gas that was captured during the dynamic phase of fermentation?
Yes, information was added to main text.
L107-135: How were the bottles filled with the gases (explain the technology) and how the final pressure in the bottles/CO2 concentration was measured (explain the measurements)? One bar, 2 bars, and 3 bars – explain the variants in more detail. Please add more details on sensory analysis, were the samples coded, how many replicates, how the assessors were trained, describe the room, etc.
Informations were adeed to main text. The water in the vessel was cooled to 5 °C. The 1 L vessel was connected to a pressure vessel with fermentation gas, then the gas from the pressure vessel was blown into the vessel with water. The gas was flowed until it had settled to a pressure of 1,2,3 bars over a long period of time. The pressure in the vessel was measured using a barometer on the pressure reducing valve.
L153-156: Which compounds are formed in the last phase of fermentation and have a negative effect on wine quality?
Thank you, volatile sulphur compounds, information was added. It will be also subject of next study.
Figure 2: You measured the concentration of volatiles per liter of CO2 gas – have you measured total gas volume produced during this fermentation and how? Or you can calculate it from the initial concentration of reducing sugars and average pressure? It would be interesting to assess what was the total loss of volatile aroma compounds during this fermentation process. There are error bars – which replicates were used for this (fermentation – I suppose not, or analysis)?
It was calculated, information was added.
L175-176: Not completely true, please be careful when discussing you results.
Thank you, it was deleted.
L194: Figure 2 – it was already stated in Line 157.
Corrected.
L195-198: Now you mention some completely different volatile aromas that are also important (ethyl acetate, isobutyl acetate, 2-phenethyl acetate), I agree. Why are they mentioned? Why they were not analyzed?
It was written very generally, but thanks for pointing it out. It is corrected.
L200. Again, please define clearly what is the final mixture of fermentation gases.
Corrected.
L199-205 & Figure 3: The concentrations you mention in the text and those on Figure 3 do not correspond at all, please revise. Please use English names for the compounds in Figure 3.
Thank you, corrected.
Paragraph 3.3. How the statistics were performed, what were the replicates in sensory analysis results?
Information was added.
L224-225 & Figures: salty vs. savoury – please adjust the text and figures attributes to match. The titles should be updates so readers can understand that fermentation gases were produced during production of wine. You use sour and then acidity – please unify as well.
Thank you, it was corrected and unified.
Authors cannot discuss the differences between waters saturated with industrially produced CO2 and mixture of purified fermentation gas since they were not statistically compared – please revise! As well, judging from very large error bars, there were no statistically significant differences between 1 bar, 2 bars and 3 bars.
Thank you, corrected and new graphs were added.
L234-236: Please support these theses with some scientific evidence (at least by referencing other works).
Information was added.
L249: This is not Figure 2.
Corrected.
Figures: Please indicate statistical significance in all figures!
Information was added.
Figure 5 title should be more informative. What is K-mixture, what is R-industrially? (K- and R-?).
Corrected. It was mistake.
Paragraph 3.5. Authors discuss differences but it is not clear which one are statistically significant. Please indicate statistical significance, as suggested above. How did you relate volatile compounds with these grades for sensory descriptors (e.g. L303-304), it should have been done more systematically, you just guessed it approximately. Please take another approach and re-assess.
Corrected.
The whole Results and discussion section should be revised, first the results should be statistically elaborated in a more systematic and accurate manner and then an appropriate discussion should follow. The current version of discussion is not at the level of Foods.
The conclusion section should also be revised, there are some parts repeated from the Introduction which should be excluded. Conclusions should contain main findings of the study (in short), future prospects and directions in this research area.
Results and discussion was revised and corrected, Conclusion was rewritten. Thank you again for your valuable comments.

Reviewer 2 Report
The objective of this research is to study the aromatic substances that are released with CO2 gas as a result of fermentation of sugar in winemaking
This topic is very appropriated as fits very well with a important concern in wine industry: to recover and reuse the carbonic gas that it is produced in fermentation. It is very well related with the need of the reduction of the carbon foot-print of wine fermentation industry. So I have found this study quite valuable.
I have some comments should be revised or probed more consistently.
a) Key worlds: wine fermentation or must fermentation, instead fermentation gas.
b) Introduction:” The novel concept of this study is the capture of fermentation gas, which may have many uses, into pressurized (with S no Z) bottles. In the wine industry, industrially produced CO2 is used as an inert gas, cooling medium or in the production of sparkling wines” I would like remark that the first one, inert gas is wide often use in the industry, much more than the others. And this fact gives to this article more importance. Wineries use gas for protect wine, and produce gas during fermentation, if they could take the gas produced to use, would be an important improvement in sustainability as well as economic. Because of that it is important to which compounds are released with CO2, positive and negative.
c) Material and methods:
· In general, explain better the system to recuperate carbonic gas. In the summary (line 10) a “new technology is mentioned”. Please explain
· 2.2. Sampling: “a sample was taken from the storage vessel where the gas had accumulated during the fermentation to analyze the final concentration of the fermentation gases”. It is supposed that this is the buffer container, as is seen in the figure, better use the same term in both text and figure. Also more details about the shape and volume of the container are needed. Is the pressure increase measured?
· 2.3. Determination of volatile compounds on fermentation gas “Our attention was focused on eight important aromatic substances that are interesting for consumers due to their aroma. These were isobutyl alcohol, isoamyl acetate, aso-92 amyl alcohol, ethyl hexanoate, 1-hexyl acetate, ethyl octanoate, ethyl decanoate and ethyl 93 dodecanoate…”. I wonder why not to focus in any negative aromas, for example, hydrogen sulfide and derivatives that are very often produced during fermentation. If we recover aroma to reintroduce in a beverage, to avoid negative aromas is very important
d) Results
· Timeline in days? Or no just in hours.
· 3.2. Determination of the content of volatile substances in the final mixture of fermentation gases. “The formation of volatile substances and their release into the fermentation gas depends on many factors, such as volatile precursors in must, yeast strain and fermentation conditions” line 193: some quote is needed.
· In general I found the results relevant and well exposed
d) Conclusions are consistent and they address the main objective of the paper. I would add here a mention about when is more produce the main aromatic compounds (correct this word), as is written in line 170“The higher concentration of hexyl acetate was observed in the earlier stage of fermentation (120 hours=5 days after inoculation with just one n), and the higher concentration of isoamyl acetate was observed later (144hours=6 days after inoculation with JUST ONE N)”
Author Response
Dear Reviewer, thank you for reviewing the article and valuable comments. We have accepted all suggested corrections.
The objective of this research is to study the aromatic substances that are released with CO2 gas as a result of fermentation of sugar in winemaking
This topic is very appropriated as fits very well with a important concern in wine industry: to recover and reuse the carbonic gas that it is produced in fermentation. It is very well related with the need of the reduction of the carbon foot-print of wine fermentation industry. So I have found this study quite valuable.
I have some comments should be revised or probed more consistently.
- Key worlds: wine fermentation or must fermentation, instead fermentation gas.
Thank you, corrected.
- Introduction:” The novel concept of this study is the capture of fermentation gas, which may have many uses, into pressurized (with S no Z) bottles. In the wine industry, industrially produced CO2 is used as an inert gas, cooling medium or in the production of sparkling wines” I would like remark that the first one, inert gas is wide often use in the industry, much more than the others. And this fact gives to this article more importance. Wineries use gas for protect wine, and produce gas during fermentation, if they could take the gas produced to use, would be an important improvement in sustainability as well as economic. Because of that it is important to which compounds are released with CO2, positive and negative.
Thank you, information was addded.
- c) Material and methods:
- In general, explain better the system to recuperate carbonic gas. In the summary (line 10) a “new technology is mentioned”. Please explain
Technology is more explained in main text according to suggestions of reviewer.
- 2.2. Sampling: “a sample was taken from the storage vessel where the gas had accumulated during the fermentation to analyze the final concentration of the fermentation gases”. It is supposed that this is the buffer container, as is seen in the figure, better use the same term in both text and figure. Also more details about the shape and volume of the container are needed. Is the pressure increase measured?
Information was added into the main text and terms were unified. Buffer tank is only for smooth running of the capture system.
- 2.3. Determination of volatile compounds on fermentation gas “Our attention was focused on eight important aromatic substances that are interesting for consumers due to their aroma. These were isobutyl alcohol, isoamyl acetate, aso-92 amyl alcohol, ethyl hexanoate, 1-hexyl acetate, ethyl octanoate, ethyl decanoate and ethyl 93 dodecanoate…”. I wonder why not to focus in any negative aromas, for example, hydrogen sulfide and derivatives that are very often produced during fermentation. If we recover aroma to reintroduce in a beverage, to avoid negative aromas is very important
This is a very useful point with which we agree, but it will be the subject of further research. These samples were taken from the dynamic phase of fermentation, when not too much H2S is formed. However, in the next study, which we are currently working on (and we already have the results), we will deal with the adsorption of H2S from the fermentation gas.
- d) Results
- Timeline in days? Or no just in hours.
Thank you, corrected.
- 3.2. Determination of the content of volatile substances in the final mixture of fermentation gases. “The formation of volatile substances and their release into the fermentation gas depends on many factors, such as volatile precursors in must, yeast strain and fermentation conditions” line 193: some quote is needed.
Citations are added at the end of the paragraph (2,8,26,32)
- In general I found the results relevant and well exposed
Thank you.
- Conclusions are consistent and they address the main objective of the paper. I would add here a mention about when is more produce the main aromatic compounds (correct this word), as is written in line 170“The higher concentration of hexyl acetate was observed in the earlier stage of fermentation (120 hours=5 days after inoculation with just one n), and the higher concentration of isoamyl acetate was observed later (144hours=6 days after inoculation with JUST ONE N)”
Based on the requests of another reviewer, the entire conclusion had to be rewritten.

Round 2
Reviewer 1 Report
The manuscript is significantly improved from the original version and authors have accepted the majority of the recommendations. However, there are still some issues that need to be resolved.
L35-37: Please edit English and write the names of the esters starting with a lowercase letter
L99-105: More details about the buffer tank and the compression unit are needed, otherwise state it is a patented device and details are part of the patent and are not reported, if this is true. If this was a regular, commercial device, please describe how it functions (physical principles) and the basic parameters, as well report the type and model.
L125: Detector voltage – please revise English in the sentence.
L141-156: Please unify this part on sensory analysis and QDSA paragraph into a single paragraph.
L305-306: Please put this in the main text and not in the Figure title.
Statistical significance (post-hoc test results) is still not indicated in the figures, it should be added and the discussion should be adjusted accordingly!
English could use some polishing.
Author Response
Dear Reviewer, thank you for reviewing the article and valuable comments. We have edited the article according to your comments. I have added basic information about the gas capture device. For more detailed information, if interested, it would be advisable to contact the manufacturer Mex directly.
L35-37: Please edit English and write the names of the esters starting with a lowercase letter
Answer: Corrected.
L99-105: More details about the buffer tank and the compression unit are needed, otherwise state it is a patented device and details are part of the patent and are not reported, if this is true. If this was a regular, commercial device, please describe how it functions (physical principles) and the basic parameters, as well report the type and model.
Answer: The gas capture device is not patented. Only for hydrogen sulfide capture technology, we made a utility model that is registered in the Industrial Property Office of the Czech Republic, which will be the topic of the next article (but it wasnt used in this article, because we captured fermentation gas only from dynamic phase of fermentation). The gas capture device was developed in cooperation of Mendel university with the Mex company, which provided us with basic information about the buffer tank and compression unit.
The buffer tank is based on the principle of an expansion tank with a rubber bag with a volume of 500 liters. The purpose of the buffer tank is the trouble-free operation of the entire capture system. It is also the place of the first accumulation of fermentation gas, which is cooled here before it enters the compression unit. After filling with fermentation gas, the fuse is deactivated at a pressure of 10 bar and the fermentation gas is released into the compression unit (Recover 6, Mex, Brno, Czech Republic), which contains a condensation (filter R24 SL, Mex, Brno, Czech Republic) and a filtration unit (filters Paricer, A0, AA, AC010, AC 500, Mex, Brno, Czech Republic). The pressure is 200 bar. This system was constructed based on the cooperation of Mendel University and the Mex company (Brno, Czech Republic).
L125: Detector voltage – please revise English in the sentence.
Answer: Corrected to: The voltage of the detector was 1.5 kV.
L141-156: Please unify this part on sensory analysis and QDSA paragraph into a single paragraph.
Answer: Corrected
L305-306: Please put this in the main text and not in the Figure title.
Answer: Corrected.
Statistical significance (post-hoc test results) is still not indicated in the figures, it should be added and the discussion should be adjusted accordingly!
Answer: Post-hoc letters were added into figure 2, 4 and 6. It was not possible to insert them into figures 3 and 5, because error bars are used for statistical expression there.
English could use some polishing.
Answer: The text was checked by a native speaker before being sent.